# miR-301a-5p Regulates *TGFB2* during Chicken Spermatogenesis

**DOI:** 10.3390/genes12111695

**Published:** 2021-10-25

**Authors:** Qixin Guo, Yong Jiang, Hao Bai, Guohong Chen, Guobin Chang

**Affiliations:** 1Jiangsu Key Laboratory for Animal Genetics, Breeding and Molecular Design, Yangzhou University, Yangzhou 225009, China; DZ120190114@yzu.edu.cn (Q.G.); jiangyong@yzu.edu.cn (Y.J.); bhowen1027@yzu.edu.cn (H.B.); ghchen2019@yzu.edu.cn (G.C.); 2Joint International Research Laboratory of Agriculture and Agri-Product Safety, The Ministry of Education of China, Institutes of Agricultural Science and Technology Development, Yangzhou University, Yangzhou 225009, China

**Keywords:** miR-301a-5p, TGFB signaling pathway, spermatogenesis, chicken

## Abstract

The process of spermatogenesis is complex and systemic, requiring the cooperation of many regulators. However, little is known about how micro RNAs (miRNAs) regulate spermatogenesis in poultry. In this study, we investigated key miRNAs and their target genes that are involved in spermatogenesis in chickens. Next-generation sequencing was conducted to determine miRNA expression profiles in five cell types: primordial germ cells (PGCs), spermatogonial stem cells (SSCs), spermatogonia (Spa), and chicken sperm. Next, we analyzed and identified several key miRNAs that regulate spermatogenesis in the four germline cell miRNA profiles. Among the enriched miRNAs, miRNA-301a-5p was the key miRNA in PGCs, SSCs, and Spa. Through reverse transcription quantitative PCR (RT-qPCR), dual-luciferase, and miRNA salience, we confirmed that miR-301a-5p binds to transforming growth factor-beta 2 (*TGFβ2*) and is involved in the transforming growth factor-beta (TGF-β) signaling pathway and germ cell development. To the best of our knowledge, this is the first demonstration of miR-301a-5p involvement in spermatogenesis by direct binding to *TGFβ2*, a key gene in the TGF-β signaling pathway. This finding contributes to the insights into the molecular mechanism through which miRNAs regulate germline cell differentiation and spermatogenesis in chickens.

## 1. Introduction

Spermatogenesis is a complex and asynchronous germ cell formation process that includes primordial germ cells that develop through several immature stages by undergoing mitosis, meiosis, and differentiation into haploid spermatozoa and mature sperm [1,2,3]. The process of spermatogenesis undergoes a well-defined order of mitotic expansion, meiotic reduction division, and Spermiogenesis [2,4] (Griswold 2016; La and Hobbs 2019). During spermatogenesis, many cell types are involved, including Sertoli cells, primordial germ cells (PGCs), spermatogonial stem cells (SSCs), spermatogonial cells (Spas), primary spermatocytes (PSCs), secondary spermatocytes (SSACs), spermatids, and spermatozoa [5,6]. PGCs, SSCs, and Spas are the key cell types in the early stages of spermatogenesis.

Recently, many studies have shown that several regulator types which include methylation, ubiquitination, acetylation, transcription factor and et.al., play major roles in germline cell development and spermatogenesis [7,8,9,10,11,12]. Small RNAs, which are a class of short non-coding RNAs, play a vital role during the entire spermatogenesis process [13,14,15]. MicroRNAs are a kind of single-stranded, non-coding small RNA that function by guiding a crucial co-factor, an AGO protein of the Argonaute family, to target genes [16]. Recent studies on spermatogenesis have demonstrated that miRNA dysregulation has been implicated in male fertility, leading to sperm abnormality and spermatogenesis disruption, suggesting that miRNAs are functionally important in the spermatogenesis process in mammalian and poultry [13,14,15,17,18]. Our previous studies also showed that miR-202-5p expressed and regulated *LIMK2* during the early spermatogonial stage in chickens [15]. Meanwhile, many miRNAs have been found in spermatozoa and have been shown to be expressed differently in high and poor fertility bulls [19,20,21,22]. Besides, a previous study showed that miRNA also regulates spermatogenesis in humans, mice, etc. In humans, MiR-133b and miR-202 are implicated in the pathophysiology of azoospermia [14].

In addition, several genes and pathways are crucial for spermatogenesis [23,24,25,26,27], including the transforming growth factor-β/Smad (TGFβ/Smad) [28], AMP-activated protein kinase (AMPK) [26,29], and mitogen-activated protein kinase (MAPK) signaling pathways [30,31]. During embryonic development, embryogenesis, and adulthood, the TGF signaling pathway regulates a variety of biological activities, including cell proliferation, differentiation, and apoptosis [32]. TGF-β2 (*TGFβ2)*, one of the three TGF-β isoforms, has been suggested to be involved in the regulation of spermatogenesis. However, few studies have reported the regulation of the TGF-β pathway in spermatogenesis by miRNAs. In the present study, we sequenced the small RNA of four cell types PGCs, SSCs, Spas, and sperm in the chicken and filtered the specific miRNAs of each cell type. Next, we want to investigate many critical miRNAs and their target genes that have been related to germ cell growth and spermatogenesis. We also want to propose an epigenetic explanation for this process.

## 2. Materials and Methods

### 2.1. Ethical Approval

All chicken and egg studies were carried out in line with the Rules for the Administration of Experimental Animals published by the Ministry of Science and Technology (Beijing, China) in 1988 (last modified in 2001). Yangzhou University’s Animal Care and Use Committee authorized the experimental protocols (YZUDWSY2017-11-07). Every attempt was made to reduce animal discomfort and suffering.

### 2.2. Flow Cytometry

PGC colonies were identified using a mouse anti-chicken c-KIT antibody (1:50) as a primary antibody and goat-anti-mouse FITC-conjugated IgM (1:50) as a secondary antibody (both from Santa Cruz Biotechnology, Santa Cruz, CA, USA). The identified PGCs were subjected to periodic acid Schiff (PAS) staining, as previously described [33]. Briefly, PGCs were fixed in 4% paraformaldehyde for 5 min, rinsed twice with 1× PBS (Hyclone, South Logan, UT, USA), immersed in the periodic acid solution for 5 min, washed with 1× PBS, immersed in Schiff’s solution for 15 min, washed twice in 1× PBS, and observed under an inverted microscope. Cells were stained for alkaline phosphatase using the 5-bromo-4-chloro-3-indolyl phosphate/nitroblue tetrazolium (BCIP/NBT) alkaline phosphatase substrate kit (Wuhan, Booster, Wuhan, China) [34]. The c-KIT and ITGA6 antibodies were used to label PGCs and SSCs, respectively, for FACS analysis performed using a FACSAria flow cytometer (BD Biosciences, San Jose, CA, USA). SaCs were sorted as previously described.

### 2.3. Data Processing

The miRNA expression data used in the present study were obtained from our previous study [11,15]. The data were analyzed for differences in miRNA expression levels during spermatogenesis. The cells for small RNA sequencing were PGCs, SSCs, Spa, and sperm. The PGCs were isolated from the gonads of chicks hatched for 5.5 days (Stage 28). The SSCs were obtained from the testes of chicks hatched for 18 days. Flow cytometry was used to extract and prepare spermatogonia single-cell suspensions from chicken testicular tissues. Chicken sperm were obtained using artificial harvesting. The Illumina HiSeq 2500 system (Illumina, San Diego, CA, USA) was used for small RNA sequencing. The clean reads of each cell type were not less than 30 M—the clean reads are shown in Table 1. The Q20 value of the base mass was greater than 90%. The raw reads used were preprocessed using FASTX. The effective reads results are shown in Table 1.

### 2.4. Key miRNA Determination and miRNA Target Gene Prediction

To identify the critical miRNAs for each stage of spermatogenesis, we first used FastQC’s FASTX-toolkit to exclude low-quality reads with Q-values of 20 and short read tags less than 18 bp. We then aligned the clean reads to the miRBase, ncRNA, and Rfam databases using the CLC Genomics Workbench. Furthermore, Venn analysis was used to evaluate the miRNA annotation in each library to identify specific miRNA expressions in each spermatogenesis-stage cell.

To predict the target genes for each key miRNA, miRDB [35], TargetScan [36], and DIANA Tools [37] were used. In addition, Venn analysis was used to filter the target genes of each miRNA.

### 2.5. miRNA Inhibition Studies and RNA Interference

The miRNA zipper, which may be utilized to cause miRNA loss-of-function, was developed to detect miRNA function during spermatogenesis (Meng et al. 2017), was used to silence miRNA and detect the effect on target miRNA genes. The LNA miRNA zippers were designed following the LNA Oligo Tools and Design Guideline (Exiqon A/S, Vedbaek, Denmark) and synthesized by GenScript (Nanjing) Co., Ltd. (Nanjing, China). The oligo sequences were (from 5′ to 3′): miR-17 zipper 5′-A+AGCACTTTGGCTACCTGCACT+GT-3′; miR-17 zipper mutation 5′-A+AGCACTATGGCTACCTGGACT+GT-3′; miR-301a-5p zipper 5′-A+TGTTGCACTACTTCTGA+CA-3′; miR-301a-5p zipper mutation 5′-A+TGTTGCGACTACTTCTGA+CA, and the negative control was a commercially available LNA product from Exiqon.

The siRNAs were designed and synthesized by Shanghai GenePharma Co., Ltd. (Shanghai, China). The siRNA 1 sequence for *TGFβ2* was sense 5′-CCCUCGACAUGGAUCAGUUTT-3′ and antisense 5′-AACUGAUCCAUGUCGAGGGTT-3′. The siRNA2 sequence for *TGFβ2* was sense 5′-CCAAGCUAUUACAGCCUUUTT-3′ and antisense 5′-AAAGGCUGUAAUAGCUUGGTT-3′. The siRNA3 sequences for *TGFβ2* were sense 5′-GCUGUACCAGGUUCUGAAATT-3′ and antisense 5′-UUUCAGAACCUGGUACAGCTT-3′. The sequences for the negative control siRNA were sense 5′-UUCUCCGAACGUGUCACGUTT-3′ and antisense 5′-ACGUGACACGUUCGGAGAATT-3′. The cells were transfected with 50 nM siRNA using Lipofectamine 2000 (Invitrogen, Waltham, MA, USA) according to the manufacturer’s instructions. The transfection mixture was replaced with DMEM/10% FBS medium 6 h later, and the cells were cultured until analysis.

### 2.6. Expression Analysis of miRNA and Genes

TRIzol reagent was used to extract total RNA (Invitrogen). The TaqManTM MicroRNA Reverse Transcription Kit (Thermo Fisher Scientific, Waltham, MA, USA) was used to produce first-strand complementary DNA of miRNAs according to the manufacturer’s instructions. The primer sequences for real-time PCR (RT-qPCR) of the miRNAs are shown in Table 2. All primer oligonucleotides were synthesized and purified by GenScript. The miRNA Universal SYBR qPCR Master Mix was produced by Vazyme Biotech Co., Ltd. (Nanjing, China). U6 was used for normalization. The QuantStudio 5 Real-Time PCR System (Applied Biosystems, Waltham, MA, USA) was used for quantitative RT-PCR. The HiScript^®^ II One-Step RT-PCR Kit (Dye Plus; Vazyme Biotech Co., Ltd.) was used for reverse transcribing RNA and quantitative. The gene primer sequences are shown in Table 2. GAPDH was amplified as an endogenous control.

### 2.7. KEGG Pathway and GO Enrichment Analysis

We performed Gene Ontology (GO) and Kyoto Encyclopedia of Genes and Genomes (KEGG) analyses on the target genes of each miRNA to identify the main functions of the genes and to reveal the miRNA-gene regulatory network, respectively. The *p*-value (*p* < 0.05) was used to determine the significance of GO term enrichment and pathway analysis. The Bonferroni method, which is a method of R/stats package, was used to adjust the *p*-value in GO enrichment and KEGG pathway analysis.

### 2.8. Construction of Recombinant Expression Vectors and the Dual-Luciferase Reporter Assay

In a pGL3 basic vector, the 3′-UTR of the *TGFB2* gene containing two gga-miR-301a-5p binding sites (1929 and 3281 bp) was subcloned downstream of the firefly luciferase reporter gene (Promega, Madison, WI, USA). Table 2 lists the primer sequences utilized in the current research. DF1 cells cultured in 24-well plates were transiently co-transfected with 400 ng luciferase vector pGL3–TGF2-3′UTR or pGL3–TGF2-3′-mUTR and either miR-301a-5p mimic or miRNA negative control for reporter experiments (miRNA-NC). As a control, 20 ng of pRL-SV40 (Promega) was co-transfected to evaluate transfection effectiveness. The Dual-Luciferase Assay System was used to conduct reporter assays 36 h after transfection (Promega).

### 2.9. Cell Culture

DF1 cells were cultured in Dulbecco’s Modified Eagle’s Medium (DMEM) supplemented with 10% fetal bovine serum (FBS) at 37 °C in a 5% CO_2_ condition with saturated humidity. DF1 cells (10^4^ cells/well) were transplanted onto 24-well plates for 1 day to confirm that the DF1 cell density reached 80%. The cells were then cultured in an incubator at 37 °C with 5% atmospheric CO_2_ and 60–70% relative humidity.

### 2.10. Statistical Analysis

All data in this research are presented as the mean standard deviation of three separate experiments. The Student’s *t*-test in the R/stats package was used to evaluate statistical significance. The statistical significance threshold was set at *p* < 0.05.

## 3. Results

### 3.1. Identification of Co-High-Expression miRNAs of Each Spermatogenesis Stage

Spermatogenesis is a dynamic and complicated cellular differentiation critical mechanism for male reproduction. miRNAs are critical gene expression regulators involved with spermatogenesis. To identify the key miRNAs in each spermatogenesis stage, we compared the miRNAs in PCGs, SSCs, and Spa with miRNA in sperm and filtered the miRNAs at log2 (fold change) ≥2 and miRNA count ≥100. In the comparison group of PGCs and sperm, we identified 156 miRNAs, 155 miRNAs in SSCs compared with sperm, and 146 miRNAs in Spa compared with sperm. Meanwhile, we identified 128 differentially expressed miRNAs in the three comparison groups (Figure 1 and Appendix A). Thereafter, we predicted the target genes of all 128 miRNAs using miRDB, DIANA Tools, and TargetScan, obtaining 1358 target genes. Furthermore, the results of GO analysis showed that almost all target genes were associated with GO entries related to cell growth, development, and cell differentiation (Appendix A). KEGG annotation results showed that almost all target genes were related to the Wnt signaling pathway, RNA degradation, and the GnRH signaling pathway (Appendix A).

### 3.2. miR-301a-5p May Affect TGFB2 Expression in Chicken

Previous studies have reported that miR-301a-5p functions as a proliferation gene in cancer and cell development. Furthermore, *TGFB2* may regulate the duration of germ cell quiescence and is required for adult spermatogenesis. Based on the predicted miRNA target genes, we found that *TGFB2* was among the target genes of miR-301a-5p. In addition, we observed that *TGFB2* and miR-301a-5p expression was negatively correlated (Figure 2a). Thus, we hypothesized that miR-301a-5p could affect *TGFB2* expression during spermatogenesis. To further confirm the effect of miR-301a-5p on *TGFB2* expression in chickens, we designed a pair comprising a miR-301a-5p zipper and miR-301a-5p zipper mutation, which have been shown to silence miRNA expression. To determine miRNA zipper function, we designed a pair comprising a miR-17 zipper and miR-17 zipper mutation and transfected it into DF1 (Figure 2b). The results showed that the silencing efficiency of the miR-17 zipper reached 72%, and that of the miR-17 zipper mutation reached 47%. Therefore, we transfected the miR-301a-5p zipper and miR-301a-5p zipper mutation into DF1 and PGCs and tested the expression of miR-301a-5p and *TGFB2* in the two cell types (Figure 2c–e). We revealed that with a decline in miR-301a-5p expression, TGFB2 expression increased. Thus, we suggest that miR-301a-5p may affect *TGFB2* expression.

### 3.3. miR-301a-5p Binds to TGFB2 Genes in Chicken

We created a miR-301a-5p mimic and inhibitor and transfected them into DF1 and PGCs, respectively, to confirm the connection between miR-301a-5p and *TGFB2*. First, we used DF1 to evaluate the efficacy of the miR-301a-5p mimic and inhibitor. The results of miR-301a-5p expression after miR-301a-5p mimic and inhibitor transfect showed that the mimic and inhibitor were available, and 65 nM for the mimic and 200 nM for the inhibitor were applicable densities for DF1 (Figure 3a,b). The same experiment was then carried out using transfected PGCs. The findings revealed a substantial difference between the PGCs-mimic, PGCs-inhibitor, and PGCs-control groups. On the one hand, the PGCs-mimic group showed 321.3 times the expression of the other two groups in miR-301a-5p, while the expression of *TGFB2* was the lowest of the three groups. The PGCs-inhibitor group, on the other hand, was 0.4 times larger than the other two groups; however, *TGFB2* had the greatest expression in all three groups (Figure 3c,d). This result suggested that the sequencing results were credible and that there was a relationship between miR-301a-5p and *TGFB2*. The dual-luciferase experiment was also used to determine the relationship between miR-301a-5p and *TGFB2*. The experimental group had lower fluorescence activity than the control group, suggesting that miR-301a-5p may inhibit *TGFB2* transcript regulation (Figure 3e).

### 3.4. miR-301a-5p Regulates the TGFB Pathway by Directly Binding to TGFB2

The above results show that miR-301a-5p may affect *TGFB2* expression by directly binding to *TGFB2*. To determine the effect of miR-301a-5p on the TGFB pathway, we first designed three siRNAs and transfected them into DF. The *TGFB2* expression results showed that siRNAs might silence *TGFB2* to 35.5 and 32%, respectively (Figure 4a). We also tested the expression of *SAMD2*, *SAMD3*, *SAMD5*, *TGFBR2*, and *TGFBR1* after siRNA-2 transfection into PGCs (Figure 4b) and observed a change in expression. Furthermore, we transfected the miR-301a-5p mimic, miR-301a-5p inhibitor, miR-301a-5p zipper, and miR-301a-5p zipper mutations into DF1 and PGCs. The aforementioned genes showed significant expression changes in the two cell types (Figure 4b,c). Thus, we propose that miR-301a-5p may regulate the TGFB pathway during spermatogenesis by directly binding to *TGFB2* (Figure 4d).

## 4. Discussion

Spermatogenesis is a complex and highly regulated process that supports the production of millions of sperm in chickens [2,3]. During spermatogenesis, PGCs are specified as germ cells and begin a process of rapid proliferation, reprogramming, and meiosis, ultimately becoming sperm [4]. More than a thousand protein-coding genes involved in spermatogenesis have been identified [25]. However, the mechanisms that mediate the expression of these spermatogenesis-related genes have not been fully elucidated. MicroRNAs (miRNAs, miR), small (~22 nucleotides) single-stranded non-coding RNAs that are critical regulators of gene post-transcriptional levels, are linked to cell proliferation, differentiation, and apoptosis [38,39,40]. Furthermore, miRNAs are involved in multiple developmental processes in many organisms, including spermatogenesis [41], and are differentially expressed in a cell-specific and step-specific manner. Several studies have shown that the miR-34 family [42,43,44], the miR-17-92 cluster [45], and other miRNAs are regulate spermatogenesis. MiR-301a-5p is an important miRNA that regulates several processes [46]. In the present study, we found that miR-301a-5p was highly expressed at each stage of spermatogenesis. This suggests that miR-301a-5p may participate in spermatogenesis. During the determination of miR-301a-5p, we observed that its expression levels gradually decreased in PGCs, SSCs, Spa, and sperm. Furthermore, using target gene prediction in three databases, we found that *TGFB2* was among the target genes of miR-301a-5p. Previous studies on the TGFb signaling system in mice have shown the TGFb negative regulator of germ cell proliferation in fetal and newborn animals and to reduce gonocyte proliferation in vitro [47,48]. TGFb, on the other hand, controls the proliferation of germ line stem cells and spermatogonia in *Drosophila* testis [49]. Previous research has shown that TGF-beta receptor isoforms and isoforms control testis development and influence steroidogenesis, cord formation, and gonocyte behavior [50,51,52,53].

In the present study, *TGFB2* expression levels gradually increased in PGCs, SSCs, Spa, and sperm. It has been suggested that miR-301a-5p negatively regulates *TGFB2* expression. Furthermore, we found that *TGFB2* expression was downregulated in the miR-301a-5p zipper and miR-301a-5p mutation zipper groups. Meanwhile, the dual-luciferase assay also showed that miR-301a-5p negatively regulated *TGFB2* through direct binding. The TGFβ signaling pathway is one of the key pathways involved in cell regulation. *TGFB2* initiates signal transduction by binding to the *TGFBR2* receptor serine/threonine kinase and transmits a signal to the Smad protein. Thus, we surmised that miR-301a-5p is involved in the TGFB signaling pathway by directly binding to *TGFB2* and participates in spermatogenesis.

## 5. Conclusions

In summary, we identified 128 miRNAs with high co-expression at each stage during spermatogenesis. Furthermore, among all high co-expression genes, we determined that miR-301a-5p may play an essential role in spermatogenesis by directly binding to *TGFB2*. In addition, miR-301a-5p may also be involved in the TGFB signaling pathway. These results provide a strong foundation for the study of azoospermia in chickens.

## Figures and Tables

**Figure 1 genes-12-01695-f001:**
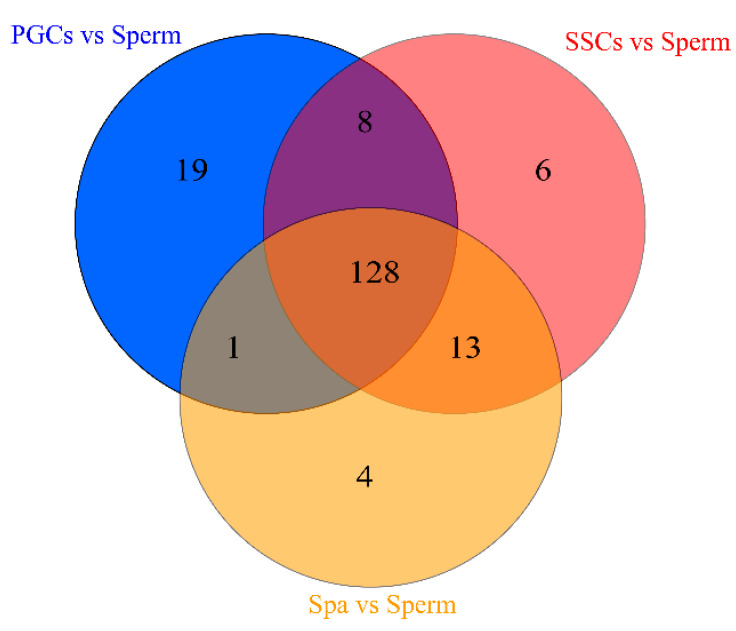
Venn analysis of the differentially expressed miRNAs in the three comparisons.

**Figure 2 genes-12-01695-f002:**
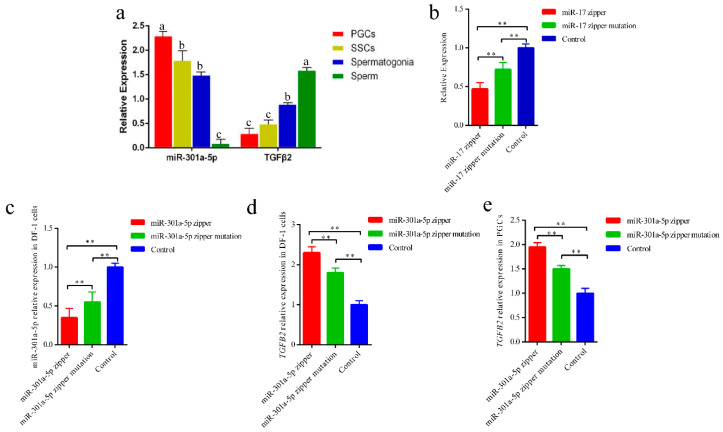
miR-301a-5p may affect *TGFB2* expression in DF-1 and PGCs. (**a**) miR-301a-5p and *TGFB2* expression during spermatogenesis; (**b**) quantitative real-time PCR analysis showing the decreased level of miR-17 in DF-1 after transfection of the miR-17 zipper and miR-17 zipper mutation; (**c**) quantitative real-time PCR analysis showing the decreased level of miR-301a-5p in DF-1 after transfection of the miR-301a-5p zipper and miR-301a-5p zipper mutation; (**d**,**e**) expression analysis showing the *TGFB2* change in DF-1 and PGCs after transfection of the miR-301a-5p zipper and miR-301a-5p zipper mutation. Data shown as mean ± SEM, ** *p* < 0.01. Different letters indicate statistically significant differences at *p* < 0.05.

**Figure 3 genes-12-01695-f003:**
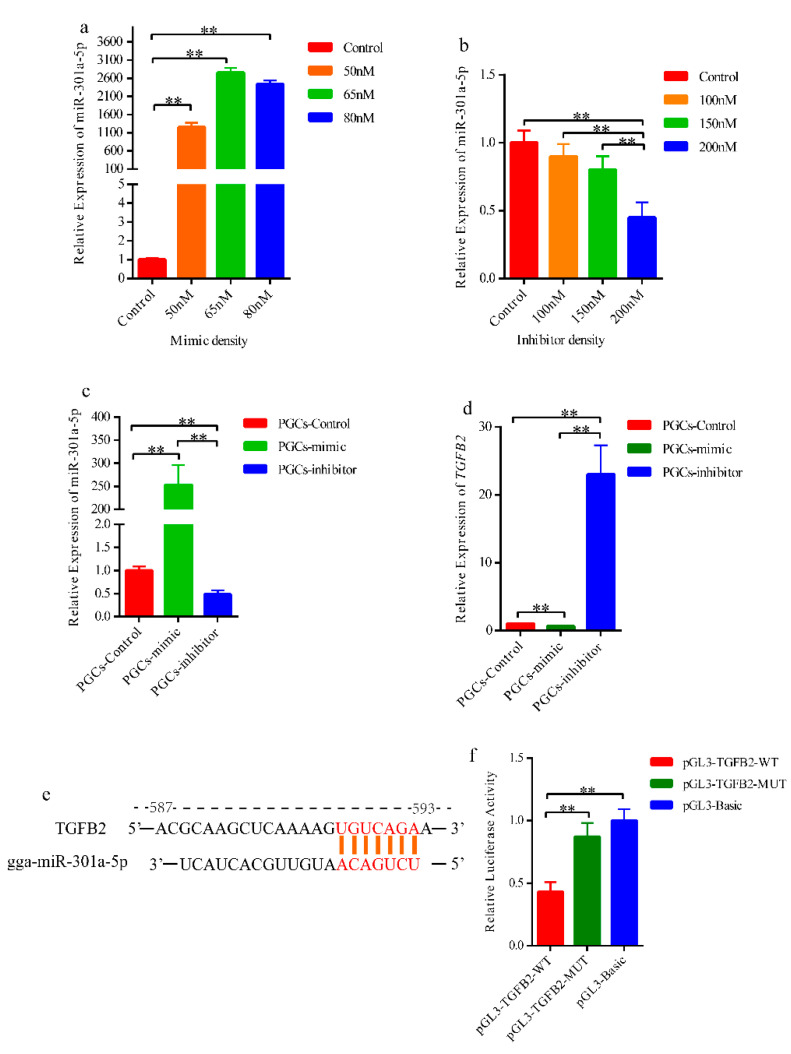
miR-301a-5p regulated *TGFB2* in DF-1 and PGCs in chicken. (**a**,**b**) Quantitative real-time PCR analysis showing the effect of transfecting different miR-301a-5p mimic and miR-301a-5p inhibitor densities in DF-1; (**c**,**d**) quantitative real-time PCR analysis showing the decreased level of miR-301a-5p and *TGFB2* in PGCs after transfection with a miR-301a-5p mimic and miR-301a-5p inhibitor; (**e**) binding site of miR-301a-5p in *TGFB2*; (**f**) results of the luciferase reporter assay revealing the relationship between miR-301a-5p and *TGFB2*. Data shown as mean ± SEM, ** *p* < 0.01.

**Figure 4 genes-12-01695-f004:**
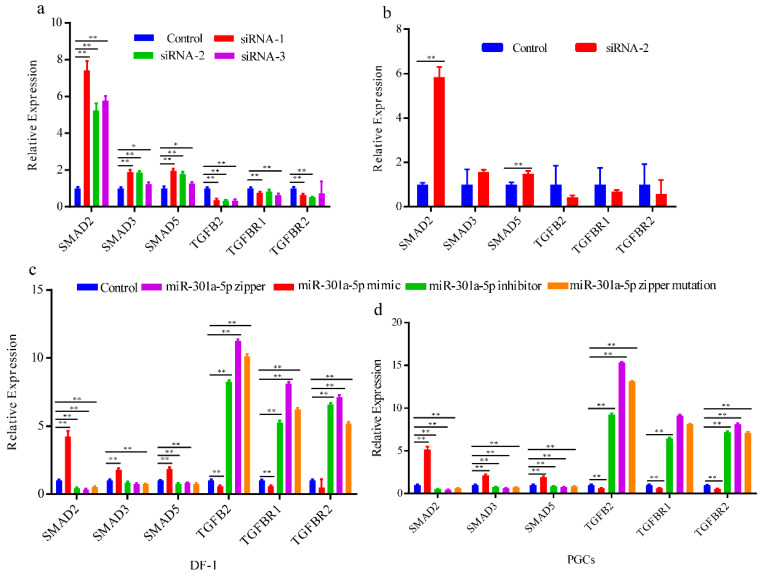
The function of miR-301a-5p in the TGFB pathway. (**a**) The expression changes of the download follow genes in the TGFB pathway in DF-1 after transfection with three siRNAs; (**b**) expression changes of the download follow genes in the TGFB pathway in PGCs after transfection with siRNA-2; (**c**,**d**) expression changes of the download follow genes in the TGFB pathway in DF-1 and PGCs after transfection with the miR-301a-5p mimic, miR-301a-5p inhibitor, miR-301a-5p zipper, and miR-301a-5p zipper mutation. Data shown as mean ± SEM, * *p* < 0.05, ** *p* < 0.01.

**Table 1 genes-12-01695-t001:** Information on small RNA sequencing data.

Samples	Clean Reads	Effective Reads	Effective Ratio
PGCs	31,361,989	30,851,348	98.37%
Spa	32,522,297	31,688,775	97.44%
Sperm	32,219,701	31,970,934	99.23%
SSC	31,757,666	31,461,002	99.07%

**Table 2 genes-12-01695-t002:** The primer of all genes which in present study.

No.	Gene Name	Primer
1	*GAPDH*	Forward: GCAGATGCAGGTGCTGAGTA
		Reverse: GACACCCATCACAAACATGG
2	*TGFβ2*	Forward: AAATGCCATCCCACCA
		Reverse: GCTCTATCCGCTGCTCC
2	*TGFβR1*	Forward: TGCGGACAACAAAGAC
		Reverse: GCCTAACTGCCAACCC
3	*TGFβR2*	Forward: GCCTACCGCACTCACA
		Reverse: TTCAATGGGCAGCAAT
4	*SMAD2*	Forward: GCCATTACCACTCAGAAC
		Reverse: TTTACGATGCGACACCT
5	*SMAD3*	Forward: GGCACATCGGAAGAGGA
		Reverse: GGTTTACAGACTGAGCCAAGA
6	*SMAD5*	Forward: TCGCCAAACAGTCCC
		Reverse: GCAACAGGCTGAACATC

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
