# Peer review of "miR-301a-5p Regulates TGFB2 during Chicken Spermatogenesis"

_genes, 2021, doi:10.3390/genes12111695_

Round 1
Reviewer 1 Report
Abstract: No comment
Introduction: Line 29 said "the process", is that means spermatogenesis? Line 30, continuation of the sentence is also said spermatogenesis. The sentence is little confusing, so if you can rewrite.
Line 36 said several regulatory types, was that meant genes? please clarify.
Line 43-46, are they about human fertility?
Line 49 said about bull fertility, but if you want to a through review, then mention human, mouse, bull, chicken all previous studies. Add 2-3 lines.
Material and methods: Line 157, heading will be cell culture, not statistical analysis.
Results: Figure 2 graph texts are not readable.
Discussion: No comment, but overall can be little more elaborated.
Conclusion: no comment.
Author Response
Response to Reviewer 1 Comments
Abstract: No comment
Introduction: Line 29 said "the process", is that means spermatogenesis? Line 30, continuation of the sentence is also said spermatogenesis. The sentence is little confusing, so if you can rewrite.
Response 1: Thanks for your kindly remind. According to your suggestion, we have recheck this sentence and rewrite the sentence to “The process of spermatogenesis undergoes a well-defined order of mitotic expansion, meiotic reduction division, and Spermiogenesis.”
Line 36 said several regulatory types, was that meant genes? please clarify.
Response 2: Thanks for your kindly suggestion. Base on your suggestion, we have added the regulatory types in this sentence.
Line 43-46, are they about human fertility?
Response 3: Thanks for your comment. Base on some reference, we found the miRNA also effect human fertility. The reference was list in follow:
Munoz, X.; Mata, A.; Bassas, L.; Larriba, S. Altered mirna signature of developing germ-cells in infertile patients relates to the severity of spermatogenic failure and persists in spermatozoa. Sci Rep 2015, 5, 17991.
Chen, X.; Li, X.; Guo, J.; Zhang, P.; Zeng, W. The roles of micrornas in regulation of mammalian spermatogenesis. J Anim Sci Biotechnol 2017, 8, 35
Line 49 said about bull fertility, but if you want to a through review, then mention human, mouse, bull, chicken all previous studies. Add 2-3 lines.
Response 4: Thanks for your comment. We have added some materials in this sentence.
Material and methods: Line 157, heading will be cell culture, not statistical analysis.
Response 5: Thanks for your comment. We have change the statistical analysis to cell culture.
Results: Figure 2 graph texts are not readable.
Response 6: Thanks for your comment. We have rewrite the graph texts.
Discussion: No comment, but overall can be little more elaborated.
Response 7: Thanks for your comments. We have added some discussion in the part of discussion.
Reviewer 2 Report
The present study attempt to analyze the cargo of miRNA in four different type of cells: PGCs, SSCs, Spa and sperm cells.
The experimental set up is clear, but there are some comments and suggestions that need to be address in the current version of the manuscript.
Major issues:
It would be necessary to include the full list of miRNAs collected and represented in the Venn (figure 1).
Discussion section is quite short. Despite the lack of specific studies, TGFbeta2 and related genes need to be discussed for their relevance in spermatogenesis and further spermiogenesis. Moreover, miR-301a-5p small RNA need to be discussed in detail.
LIN 82. Please, describe in detail the flow cytometry isolation process.
Minor issues:
LIN 62-66. This sentence is ought to be included in the result section or the discussion more than in the introduction. Please, define the hypothesis and the aim of the experiment instead of results.
LIN 79, Spa or SPCs (as line 63) Please, check the entire manuscript for inconsistencies.
LIN 134. Please, include the primers or commercial primers for GAPDH.
LIN 146. P-value or q-values, being FDR correction.
LIN 157. Please, change the title of this section 2.8.
In figure 4, where statistical analysis are available?
Author Response
Response to Reviewer 2 Comments
The present study attempt to analyze the cargo of miRNA in four different type of cells: PGCs, SSCs, Spa and sperm cells.
The experimental set up is clear, but there are some comments and suggestions that need to be address in the current version of the manuscript.
Major issues:
It would be necessary to include the full list of miRNAs collected and represented in the Venn (figure 1).
Response 1: Thanks for your comments. We have added some discussion in the part of discussion.
Discussion section is quite short. Despite the lack of specific studies, TGFbeta2 and related genes need to be discussed for their relevance in spermatogenesis and further spermiogenesis. Moreover, miR-301a-5p small RNA need to be discussed in detail.
Response 2: Thanks for your kind comments. We have added some discussion of TGFbeta2 and related genes in the part of discussion.
LIN 82. Please, describe in detail the flow cytometry isolation process.
Response 3: Thanks for your kind comments. We have added detail the flow cytometry isolation process in the part of methods.
Minor issues:
LIN 62-66. This sentence is ought to be included in the result section or the discussion more than in the introduction. Please, define the hypothesis and the aim of the experiment instead of results.
Response 4: Thanks for your kind comments. We have rewrite this sentence.
LIN 79, Spa or SPCs (as line 63) Please, check the entire manuscript for inconsistencies.
Response 5: Thanks for your kind comments. We have recheck the SPCs to Spa in all manuscript.
LIN 134. Please, include the primers or commercial primers for GAPDH.
Response 6: Thanks for your kind comments. We have added the primer of GAPDH to Table 2.
LIN 146. P-value or q-values, being FDR correction.
Response 7: Thanks for your kind comments. We have rewrite this sentence.
LIN 157. Please, change the title of this section 2.8.
Response 8: Thanks for your kind comments. We have rewrite the section 2.8.
In figure 4, where statistical analysis are available?
Response 9: Thanks for your comments, we have added statistical analysis in figure 4.
Discussion: No comment, but overall can be little more elaborated.
Response 7: Thanks for your comments. We have added some discussion in the part of discussion.

Round 2
Reviewer 1 Report
Introduction: “Besides, previous study the miRNA also regulates the spermatogenesis in human, mice and so on.”, Line 53-55. Please change the sentence construction. If not possible then remove it.
Material and methods: Please change “TGFB2” to “TGFβ2”, line 151 or use any one of it in the whole text.
Otherwise it is good now.